# Downscaling precipitation over High Mountain Asia using Multi-Fidelity Gaussian Processes: Improved estimates from ERA5

Kenza Tazi[1,2], Andrew Orr[1], Javier Hernandez-González[3], Scott Hosking[1,4], and Richard E. Turner[2]

[1]British Antarctic Survey, Cambridge, UK
[2]Department of Engineering, University of Cambridge, Cambridge, UK
[3]Microsoft Research, Cambridge, UK
[4]The Alan Turing Institute, London, UK

**Correspondence:** Kenza Tazi (kt484@cam.ac.uk)

**Abstract.** The rivers of High Mountain Asia provide freshwater to around 1.9 billion people. However, precipitation, the main driver of river flow, is still poorly understood due to limited *in situ* measurements in this area. Existing tools to interpolate these measurements or downscale and bias-correct precipitation models have several limitations. To overcome these challenges, this paper uses a probabilistic machine learning approach called Multi-Fidelity Gaussian Processes (MFGPs) to downscale

ERA5 climate reanalysis. The method is first validated by downscaling ERA5 precipitation data over data-rich Europe and then data-sparse upper Beas and Sutlej River basins in the Himalayas. We find that MFGPs are simpler to implement and more applicable to smaller datasets than other state-of-the-art machine learning methods. MFGPs are also able to quantify and narrow the uncertainty associated with the precipitation estimates, which is especially needed over ungauged areas, and can be used to estimate the likelihood of extreme events that lead to floods or droughts. Over the upper Beas and Sutlej River basins,

the precipitation estimates from the MFGP model are similar to or more accurate than available gridded precipitation products (APHRODITE, TRMM, CRU TS, bias-corrected WRF). The MFGP model and APHRODITE annual mean precipitation estimates generally agree with each other for this region with the MFGP model predicting slightly higher average precipitation and variance. However, more significant spatial deviations between the MFGP model and APHRODITE over this region appear during the summer monsoon. The MFGP model also presents a more effective resolution, generating more structure at finer

spatial scales than ERA5 and APHRODITE. MFGP precipitation estimates for the upper Beas and Sutlej basins between 1980 and 2012 at a 0.0625° resolution (approx. 7 km) are jointly published with this paper.

## 1 Introduction

High Mountain Asia underpins the water security of around 1.9 billion people, supplying them with fresh water for agriculture, energy, industry and domestic usage via Asia's largest rivers (Wester et al., 2019; Immerzeel et al., 2020; Orr et al., 2022).

In this area, precipitation drives river flow either directly through rain or, indirectly, by depositing snow reserves that are eventually released through glacier and snow melt (Immerzeel et al., 2020). Precipitation over High Mountain Asia is mainly influenced by two large scale atmospheric patterns: the Indian Summer Monsoon and Western Disturbances, which dominate

in the boreal winter (Bookhagen and Burbank, 2010; Palazzi et al., 2013; Dimri et al., 2015). On a local scale, precipitation over High Mountain Asia is characterised by large variances across relatively small distances of 1 to 10 km due to the region's complex topography (Anders et al., 2006; Bookhagen and Burbank, 2006; Karki et al., 2017; Bookhagen and Burbank, 2010; Sigdel and Ma, 2017; Orr et al., 2017; Bannister et al., 2019). However, the spatiotemporal distribution of precipitation in this area is comparatively poorly understood (Singh et al., 2015; Dahri et al., 2021a; Orr et al., 2022).

Knowledge of precipitation patterns in High Mountain Asia is principally constrained by limited observations. Only a small number of *in situ* precipitation observations exist in this region, with most gauge stations placed in unrepresentative locations (below 2000 m a.s.l.) (Winiger et al., 2005; Salzmann et al., 2014; Immerzeel et al., 2015; Duan et al., 2015; Bhardwaj et al., 2017; Krishnan et al., 2019). Indirect observations through satellites are available but struggle to capture the distribution differences between valleys and ridges, as well as short-lived extreme events. Furthermore, satellites often confuse precipitation with ice and snow at the surface level. This leads to remote sensing products generally underestimating precipitation in mountainous areas (Yin et al., 2008; Andermann et al., 2011). These obstacles mean that many physical relationships, such as between precipitation rates and elevation, are not well understood in High Mountain Asia (Dahri et al., 2016). This in turn adversely affects tools to interpolate or combine precipitation measurements to create gridded precipitation products (Meng et al., 2014; Bhardwaj et al., 2017; Hussain et al., 2017; Ji et al., 2020). As a result, interpolation-based products such as APHRODITE (Yatagai et al., 2012) tend to underestimate precipitation at high altitudes (Immerzeel et al., 2015; Li et al., 2017). Furthermore, such gridded products often have no uncertainty estimates.

In addition to interpolation-based products, outputs from regional climate models (RCMs) can also be used to estimate precipitation over High Mountain Asia (Maussion et al., 2014; Norris et al., 2017, 2019; Orr et al., 2017; Norris et al., 2020). However, these physical models are computationally expensive, lack error estimates, generate large model-dependent uncertainty (Hawkins and Sutton, 2009), and are generally not well-optimised for mountainous regions (Cannon et al., 2017; Norris et al., 2017, 2019; Orr et al., 2017). For example, the ensemble of RCMs from the Coordinated Regional Climate Downscaling Experiment (CORDEX) for South Asia regularly overestimates historical precipitation over High Mountain Asia by over 100% for both winter and summer (Sanjay et al., 2017). RCM precipitation outputs therefore typically need to be bias-corrected before use in this region (Maussion et al., 2014; Collier and Immerzeel, 2015; Bannister et al., 2019; Potter et al., 2022).

Climate reanalysis products offer an alternative for estimating precipitation by combining outputs from short-range forecast models with observations through data assimilation. These products often struggle to accurately represent precipitation over data-sparse areas or times, including High Mountain Asia (Dahri et al., 2016; Palazzi et al., 2013). ERA5 climate reanalysis (Hersbach et al., 2020), although generally exhibiting a wet bias for High Mountain Asia, provides relatively accurate precipitation estimates in terms of amounts, seasonality, and variability, from daily to multi-annual scale compared to other reanalysis and RCM products (Chen et al., 2021; Kumar et al., 2021).

Altogether, precipitation products over High Mountain Asia are often contradictory and lack consensus (Palazzi et al., 2013; Bannister et al., 2019). These discrepancies further complicate our understanding and leave room for doubt in any given

prediction or estimate. As precipitation is the main driver of hydrological models (Meng et al., 2014; Remesan and Holman, 2015; Wulf et al., 2016), improving precipitation estimates is key to a better representation of the spatial and temporal dynamics of hydrological processes. These improved estimates can in turn help us better understand, predict and mitigate extreme events such as droughts, floods and landslides (Ji et al., 2020; Dahri et al., 2021b; Schreiner-McGraw and Ajami, 2022). Present day precipitation estimates also underpin the accuracy of future precipitation predictions (Panday et al., 2015; Sanjay et al., 2017).

Traditional and state-of-the-art statistical downscaling techniques are used to address these problems but present their own issues. For High Mountain Asia, downscaling models often assume simplistic relationships, e.g. a linear correlation between precipitation and elevation, and focus on single basins (Dahri et al., 2016; Bannister et al., 2019; Libertino et al., 2018). New research is making the most of machine learning tools to downscale precipitation products (Yadav et al., 2024; Gerlitz et al., 2014), allowing researchers to model more complex spatiotemporal precipitation distributions and generate products over larger areas and longer time periods (Ahmed et al., 2020; Ning et al., 2016; Mei et al., 2020; Sun et al., 2022). These studies are also using machine learning corrected precipitation directly as inputs to hydrological models (Sun et al., 2022; Xiang et al., 2024) and applying machine learning methods to merge precipitation data from multiple sources to improve prediction robustness in ungauged areas (Lyu and Yong, 2024; Xiang et al., 2024; Zhang et al., 2021).

However, these downscaling methods generally struggle to simultaneously solve the following problems: 1) capturing extreme values and spatiotemporal structure, 2) generalising to multiple locations, 3) predicting at arbitrary locations, 4) overcoming gridding biases and 5) working effectively with sparse and 'small' datasets (King et al., 2013; Maraun and Widmann, 2018; Baño-Medina et al., 2020; ?; Andersson et al., 2023). We propose Multi-Fidelity Gaussian Processes (MFGPs) as an alternative to other statistical downscaling and bias-correction methods. Using MFGPs, precipitation data from multiple sources can be combined to overcome these challenges and increase the accuracy and effective resolution of precipitation predictions over topographically complex areas, especially over ungauged locations. Most importantly, the probabilistic nature of MFGPs provides a principled way of quantifying uncertainty and the likelihood of extreme precipitation events.

This study focuses on applying MFGPs to downscale ERA5 monthly precipitation estimates in the data-sparse upper Beas and Sutlej River basins in the Himalayas. The Beas and Sutlej are two main tributaries of the Indus River. The study area, shown in Figure 1, serves as a pilot study for High Mountain Asia. The paper is structured as follows. Gaussian Processes (GPs) and MFGPs are first introduced in Section 2. The methodology and datasets used are presented in Section 3. The MFGPs are then evaluated by downscaling ERA5 precipitation, first over a data-rich region (Europe) and then over a subset of the upper Beas and Sutlej basins, in Section 4. The MFGP framework is then applied to the whole of the upper Beas and Sutlej basins and compared with precipitation dataset benchmarks including APHRODITE in Section 5. Finally, the limitations of this approach and further work are discussed in Section 6.

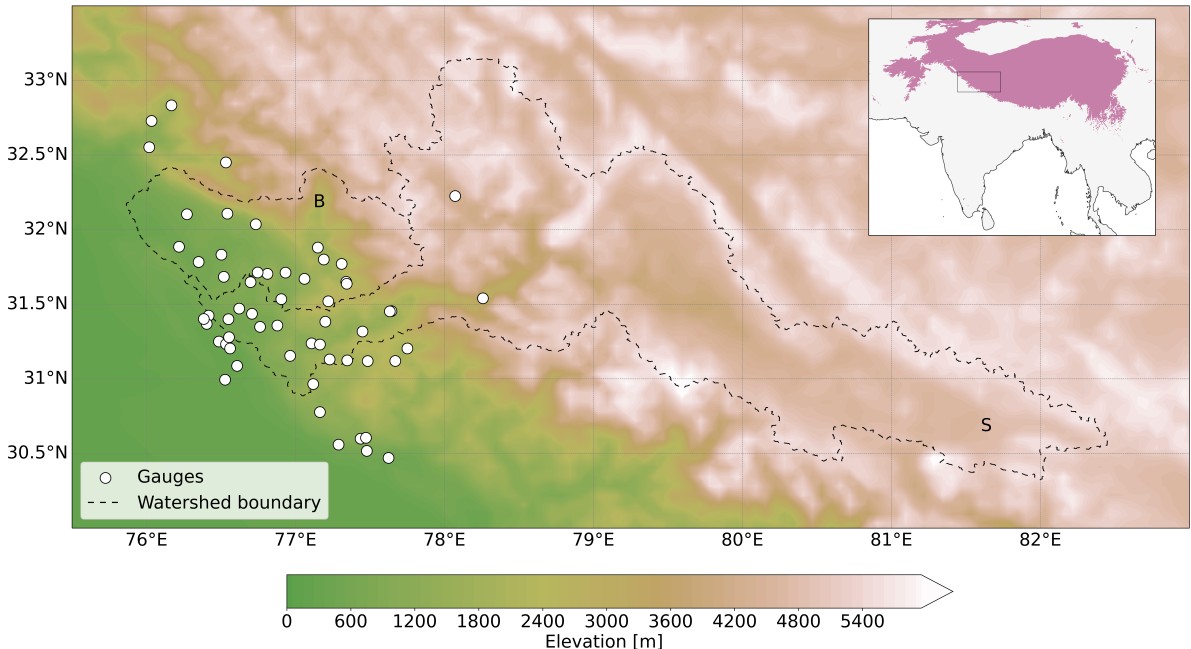

**Figure 1.** Elevation map of the upper Beas and Sutlej River basins with gauge locations represented by white circles. The dashed line represents the watershed boundaries, with the letter 'B' denoting the upper Beas basin and 'S' the upper Sutlej basin. Only three gauge stations are located above 2000 m. The inset shows the watersheds' location with respect to High Mountain Asia, with areas above 2000 m a.s.l. highlighted in purple.

## 2  Multi-Fidelity Gaussian Processes

### 2.1  Gaussian Processes

Consider the set of observations $\boldsymbol{x}_i, y_i$ with $i = \{1, ..., N\}$, $\boldsymbol{x}_i \in \mathbb{R}^D$ and $y_i \in \mathbb{R}$ where $N$ is the number of data points and $D$ the number of observation dimensions. In this paper, $\boldsymbol{x}_i$ represents a vector with the date, coordinates and elevation of the observation and $y_i$ is its monthly precipitation value. These observations are generated by a function $f$:

$$y_i = f(\boldsymbol{x}_i) + \epsilon_i \tag{1}$$

where $\epsilon_i$ is the noise term and is assumed to be distributed normally with a mean of zero and standard deviation $\sigma_n$, i.e., $\epsilon_i \sim \mathcal{N}(0; \sigma_n^2)$. The function $f$ can be modelled with a Gaussian Process (GP). We refer the reader to Rasmussen et al. (2006) for an introduction to GPs and follow their notation in this presentation. A GP is a stochastic process where any finite collection of its random variables is distributed according to a multivariate normal distribution. Similarly to a multivariate normal distribution, a GP is defined by a mean function $\mu(\boldsymbol{x}, \boldsymbol{\theta}_\mu)$ and covariance or kernel function $k(\boldsymbol{x}, \boldsymbol{x}'; \boldsymbol{\theta}_k)$:

$$f(\boldsymbol{x}) \sim GP(\mu(\boldsymbol{x}; \boldsymbol{\theta}_\mu), k(\boldsymbol{x}, \boldsymbol{x}'; \boldsymbol{\theta}_k)) \tag{2}$$

where $\boldsymbol{x}$ is the input vector to predict at, $\boldsymbol{x}'$ is another arbitrary input location, and $\boldsymbol{\theta}_\mu$ and $\boldsymbol{\theta}_k$ represent the hyperparameters of the mean and covariance functions respectively. The hyperparameters are the parameters of the model that can either be set manually or optimised. Going forward the hyperparameters will be referred to jointly as $\boldsymbol{\theta}$. The covariance function $k(\boldsymbol{x}, \boldsymbol{x}'; \boldsymbol{\theta}_k)$ strongly underpins the GP model. It captures the correlation of the outputs given the inputs encoding properties such as smoothness and periodicity. If the covariance function is stationary, the correlation depends only on the distance between $\boldsymbol{x}$ and $\boldsymbol{x}'$.

As the output of a GP for a single point is a probability distribution, the GP output over many points can be interpreted as a probability distribution over functions. Predictions at new input locations can therefore be calculated using Bayes' theorem. This is also known as the model being 'fit' to the data or 'training' the model with the data. If $A$ represents the GP's functions and $B$ the data, Bayesian inference can be written as:

$$p(A|B,C) = \frac{p(A|C)p(B|A,C)}{p(B|C)} \quad \text{where} \quad A = f(\cdot), \quad B = \{\boldsymbol{x}_i, y_i\}_{i=1}^N, \quad C = \boldsymbol{\theta} \tag{3}$$

where $p(A|B,C)$ is the probability distribution of $A$ conditional on $B$ and $C$ with all other distributions defined analogously. This can be seen as the system $A$ being updated using new information $B$. $p(A|C)$ is therefore known as the prior distribution and $p(A|B,C)$ as the posterior distribution. The posterior distribution is is therefore a principled way to define the uncertainty of the model and is therefore estimating the probabilities of extreme values. $p(B|A,C)$ is the probability of the observations $B$ occurring given the state of system $A$ with hyperparameters $C$ and is known as the likelihood. $p(B|C)$ is known as the marginal likelihood and is the probability density of the observations given the hyperparameters. This distribution is calculated by integrating or 'marginalising' over all the values of $f$, i.e. going from $p(B|A,C)$ to $p(B|C)$.

GPs therefore are non-parametric. Instead of optimising over finite set of parameters, e.g. weights of a random forest or neural network, GPs are optimised over functions. Consequently, GPs are more expressive in how they fit the data compared to traditional regression or classification models, i.e. they can be used to model complex relationships between the data. GPs are also more robust to overfitting because rather than optimise a specific function, it integrates over all potential ones (Rasmussen et al., 2006).

Practically, the mean function $\mu(\boldsymbol{x}; \boldsymbol{\theta}_\mu)$, the covariance function $k(\boldsymbol{x}, \boldsymbol{x}'; \boldsymbol{\theta}_k)$ and the prior distribution are built from a set of standard functions that encode different assumptions. In particular, the covariance matrix is usually designed by multiplying or adding standard kernel functions together (Rasmussen et al., 2006; Duvenaud et al., 2013). The covariance function makes GPs well suited for highly-correlated geophysical datasets and quantifying uncertainty in absence of data. However these benefits come at a cost, the computational complexity of GPs scales cubically with the number of data points. This scaling is an issue in large data regimes but can be addressed by low-rank approximations and inducing points (Liu et al., 2020; Tazi et al., 2023).

## 2.2 Multi-Fidelity Gaussian Processes

The fidelity of a dataset can be defined as a combination of the data's precision and accuracy. The most accurate set of observations with the highest resolution are referred to as the high-fidelity data. Less accurate and coarse observations or

simulation data are denoted as low-fidelity data. In many cases, high-fidelity observations can be expensive to produce whereas low-fidelity observations are usually more inexpensive and therefore more numerous. A multi-fidelity model combines low-fidelity datasets with the more accurate, but limited, observations in order to predict the high-fidelity output more effectively. Datasets of different fidelities can be combined using GPs, where the output of a first GP is used as the input to the next and so forth. For a Multi-Fidelity Gaussian Process (MFGP), each layer of the model represents a different level of fidelity, starting from the lowest and moving towards the highest fidelity.

Consider $s$ fidelity levels each corresponding to different datasets, e.g., climate reanalysis, gauge station measurements, etc. Each fidelity is made up of observations $\boldsymbol{Y}_t$ at locations $\boldsymbol{X}_t \subseteq \mathbb{R}^D$ where $t = 1, \ldots, s$. The observations $\boldsymbol{Y}_s$ denote the outputs of the most accurate and expensive function to evaluate $f_s$, whereas $\boldsymbol{Y}_1$ is the outputs of the cheapest and least accurate function $f_1$. The highest-fidelity data are assumed to be sampled from the 'true' distribution of the target function. The observations at level $t$ can be generated by a function $f_t$:

$$Y_{t,i} = f_t(\boldsymbol{X}_{t,i}) + \epsilon_{t,i} \tag{4}$$

where $\epsilon_{t,i}$ is the noise term.

One choice for this function is given by Le Gratiet and Garnier (2014). The approach requires two assumptions. First, the relationship between the fidelities is assumed to be linear. Second, the model follows strict hierarchical sampling rules where the fidelity levels have nested training sets. The high-fidelity locations must be contained with the domain of the lower-fidelity. The lowest-fidelity data must therefore have the largest domain, the second fidelity must have the second-largest domain and so forth. From these assumptions, the function $f_t$ is defined as:

$$f_t(\boldsymbol{X}_t) = \rho_t f_{t-1}(\boldsymbol{X}_t) + f_{\mathrm{err}}(\boldsymbol{X}_t). \tag{5}$$

The function $f_t$ is the high-fidelity GP as modelled by the scaled sum of of two functions $f_{t-1}$ and $f_{\mathrm{err}}$. The function $f_{t-1}$ is a GP modelling the outputs of the lower-fidelity function and is scaled by $\rho_t$, a scalar indicating the magnitude of the correlation to the high-fidelity data. The function $f_{\mathrm{err}}$ is another GP that models the bias between the two fidelity levels. The scaling factor $\rho_t$ is defined as:

$$\rho_t(\boldsymbol{X}_t) = \frac{\mathrm{cov}(f_t(\boldsymbol{X}_t), f_{t-1}(\boldsymbol{X}_t))}{\mathrm{var}(f_{t-1}(\boldsymbol{X}_t))} \tag{6}$$

where cov is the covariance and var is the variance. Model inference, including the propagation of the mean and standard deviation through different fidelity levels, is discussed in Appendix A. Figure 2 illustrates the MFGP framework for a pedagogical example using two toy datasets.

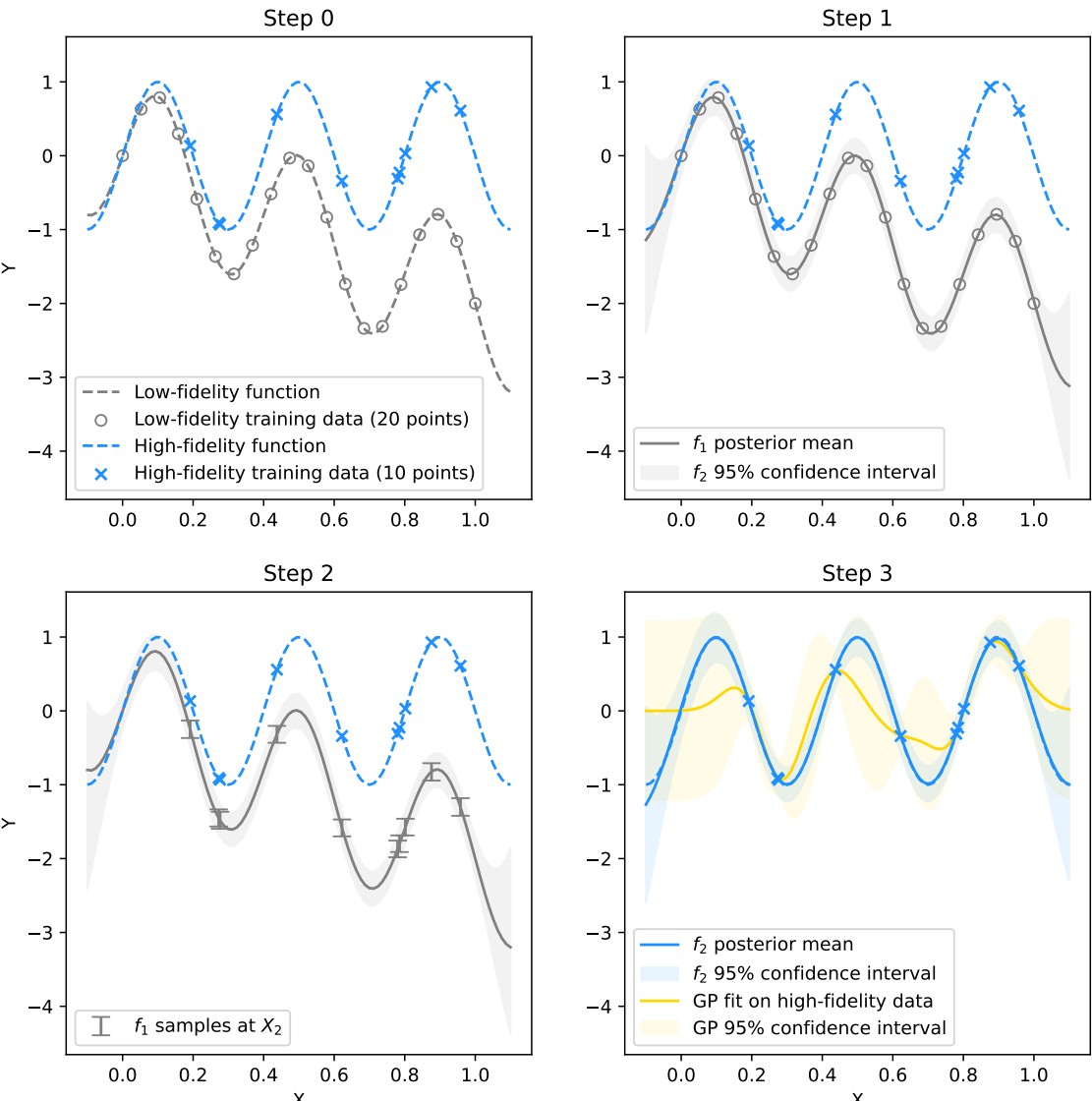

**Figure 2.** One dimensional pedagogical example of a MFGP model. The low-fidelity dataset is first contrasted with the high-fidelity data (Step 0). The high-fidelity data is more sparse but has a higher resolution than the low-fidelity data, and is also nested within the low-fidelity input domain. The first GP $f_1$ is constrained by the lowest-fidelity observations $\boldsymbol{Y}_1(\boldsymbol{X}_1)$ (Step 1). Function $f_1$ is visualised through its posterior distribution mean (grey continuous line) and its 95% confidence interval (grey shaded area) and can be used to make predictions at new locations. Samples from $f_1$ at $\boldsymbol{X}_2$ (Step 2) and the observations from the second fidelity $\boldsymbol{Y}_2(\boldsymbol{X}_2)$, are then used as the inputs to the second GP $f_2$ (Step 3). The final panel also shows the output of simple GP fit to the high-fidelity data only. The simple GP model fails to capture the underlying high-fidelity function and produces a more poorly constrained posterior distribution.

## 3 Method and datasets

### 3.1 Method overview

In this study we use the MFGP framework to combine two datasets of different fidelities: high-fidelity gauge measurements, which are accurate but sparse, and climate reanalysis data, which are complete but more biased. In this way, the MFGP is applied to downscale and bias-correct monthly reanalysis precipitation data using precipitation gauge measurements. Time, latitude, longitude and elevation are used as input variables. The datasets used to train the MFGPs and make predictions from the model are described in Section 3.2. The MFGP framework is validated using subsets of European station data and then upper Beas and Sutlej gauge data. MFGP is first applied to Europe in order to ascertain the performance of the model on an area with less sparse gauge data and more homogeneous spatial distribution of precipitation before applying it to the more challenging upper Beas and Sutlej regions. A MFGP model is then trained using all the gauges in upper Beas and Sutlej basins and compared to other benchmark datasets. The benchmark datasets, their advantages and their limitations are presented in Section 3.3.

### 3.2 Training and prediction datasets

The datasets used to train the MFGP model include the VALUE gauge measurements over Europe (high-fidelity), the Beas and Sutlej gauge measurements (high-fidelity), and ERA5 (low-fidelity). The digital elevation model is also presented and is used to make the high resolution precipitation estimates over the upper Beas and Sutlej basins.

**VALUE gauge measurements**. The European station measurements are taken from the VALUE downscaling experiment (Gutiérrez et al., 2019). The dataset features daily precipitation at 86 stations across Europe between 1979 and 2019 . These stations are representative of different climatic regimes over the European continent including mountainous environments. The daily data is re-sampled to a monthly temporal resolution.

**Beas and Sutlej gauge measurements**. The upper Beas and Sutlej basins are chosen as the study region as they offer comparatively data-rich locations for High Mountain Asia (Wulf et al., 2016; Bannister et al., 2019). The dataset from Bannister et al. (2019) with additional quality control is used. The dataset is made up of 58 stations with 46 within the upper Beas and Sutlej basins and measurements between January 1980 and April 2013. The 23 stations run by the Bhakra Beas Management Board measure rainfall and snow water equivalent. The remaining 35 stations are run by the Indian Meteorological Department and only record rainfall. This is not problematic as all these stations are below the snow line in this area (Lund et al., 2020). The precipitation observations are daily but have missing values with gaps of several years for most locations. The stations cover less than half of the study area as seen in Figure 1. With station altitudes ranging from 284 m to 3639 m a.s.l. and a median altitude of 935 m a.s.l., most stations are not representative of the combined watersheds which together have a median elevation of approximately 4700 m a.s.l. The data is resampled from daily to monthly averages.

**ERA5**. The 5[th] ECMWF Reanalysis (ERA5) (Hersbach et al., 2020) is used to train the low fidely GPs of the MFGP model. ERA5 runs from 1950 to the present day over 0.25° by 0.25° grid and assimilates data from a large number of sources. ERA5's global spatial coverage and long temporal range make it an attractive dataset. It is also easily accessible and straightforward to update. The monthly total precipitation variable is used in the following experiments. Elevation values are derived from ERA5's geopotential variable.

**GMTED2010**. The 2010 global multi-resolution terrain elevation data (GMTED2010) is a digital elevation model computed from 11 satellite data sources (Danielson and Gesch, 2011). The model provides elevation products at three separate resolutions of 30 arc-seconds (approx. 1 km), 15 arc-seconds (approx. 500 m), and 7.5 arc-seconds (approx. 250 m) with global land coverage from 84° N to 56° S for most products. In this paper, a resampled version of GMTED2010 at 0.0625° resolution from the European Space Agency's Tropospheric Monitoring Instrument team (TROPOMI, 2019).

## 3.3 Benchmark datasets

Precipitation estimates using the MFGP framework are compared against the following precipitation benchmark datasets: bias-corrected WRF, APHRODITE, TRMM, and CRU TS.

**Bias-corrected WRF**. The bias-corrected WRF output is a product that was specifically developed for the upper Beas and Sutlej basins by Bannister et al. (2019). Here, version 3.8.1 of the WRF model (Skamarock et al., 2008) was used to dynamically downscale ERA-Interim reanalysis data (Dee et al., 2011) to a grid spacing of 5 km from 1980 to 2012. The precipitation outputs from the model were then bias-corrected using the *in situ* observations described above, using a power transformation method proposed by Leander and Buishand (2007).

**APHRODITE**. The second benchmark is the Asian Precipitation-Highly Resolved Observational Data Integration Towards Evaluation of water resources or (APHRODITE Yatagai et al., 2012). APHRODITE data ranges from 1951 to 2015 with a maximum spatial resolution. The interpolation scheme uses nearby precipitation gauges, slope and a correlation distance lookup table. In the paper, we use the APHRO_V1101 gridded precipitation product which was specifically developed for monsoon Asia. Overall, APHRODITE has one of the best spatiotemporal coverage of gridded precipitation products over High Mountain Asia. It is also one of the most studied and accurate products for the region (Dimri, 2021). However, the interpolation scheme underestimates precipitation at high altitudes and suffers from spatially heterogeneous biases when compared to *in situ* observations. These biases pose critical limitations for high-precision hydrological studies (Ji et al., 2020; Bhardwaj et al., 2017; Hussain et al., 2017).

**TRMM**. The Tropical Rainfall Measuring Mission (TRMM) is a satellite mission that was launched at the end of 1997 and remained active until 2014. TRMM provides good spatial coverage over High Mountain Asia, although several studies have shown that the relatively coarse resolution of its products is unable to capture distribution differences between valleys and ridges (Shukla et al., 2019; Andermann et al., 2011; Yin et al., 2008). Additionally, its relatively poor temporal coverage (only a few overpasses per day) also contributes to extreme precipitation events not being captured. Here we use TRMM_3B43 data, which

is a monthly 0.25° resolution Level 3 precipitation product where radar and radiometer measurements have been converted to precipitation values and the results have been calibrated against ground measurements (Japan Aerospace Exploration Agency, 2018). However, the calibration sites are not in High Mountain Asia.

**CRU TS**. The final benchmark is the high-resolution Climatic Research Unit global climate Time Series dataset (CRU TS v4.05 Harris et al., 2020). This gridded dataset uses an angular-distance weighting interpolation of *in situ* observations between 1901 and 2020. This resulting product has a 0.5° resolution and was chosen as a baseline given its coarser resolution and global scope.

## 4 Model validation

### 4.1 Experimental setup

#### 4.1.1 Validation scheme

The MFGP model is evaluated from 2000 and 2004 over both Europe and the upper Beas and Sutlej basins. This time period represents the time with the largest number of active stations in the upper Beas and Sutlej basins. For both regions, the MFGP model is tested using fivefold cross-validation. This means the data are first separated into five groups or folds and five separate models are trained on different permutations of four groups and tested on the fifth. Cross-validation is therefore a useful way to estimate how the model will perform in practice when it is asked to predict at arbitrary locations far way from its training distribution. The groups are determined via k-means clustering on the station locations. To make the cross-validation clusters even in size, only the seven closest stations to the cluster centres are kept. The cluster downsizing also increases the spatial independence between folds. The folds for both regions are shown in Figure 3.

Different variants of this cross-validation method are used to evaluate the MFGP model. First, we setup a 'data-rich' experiment over Europe. In this case, all the available stations except the test fold stations are used to train the model. For example, when evaluating the model on Fold 1 (Figure 3 a), blue markers), the model is trained on the other folds and the grey stations. In this setting, the model therefore has access to more data including data that are climatically similar to where the model is evaluated. We then modify the experiment to create a 'data-sparse' setting over Europe. In this case, we train only the data in the training folds and test on the excluded group. The 'data-sparse' scheme is repeated for the evaluation over the upper Beas and Sutlej basins. This progressive reduction in data should help compare the impact of the data sparsity on the MFGP model against that of complex spatiotemporal precipitation distribution in the upper Beas and Sutlej basins.

#### 4.1.2 Data transformation

The probability distribution function of monthly precipitation is not Gaussian but usually follows a log-normal distribution. However, as the GP posterior distribution is constrained to be normal, making the marginal distribution more normal can therefore help with inference. For this reason, the precipitation data are transformed using a Box-Cox function $g_\lambda$ fit to the

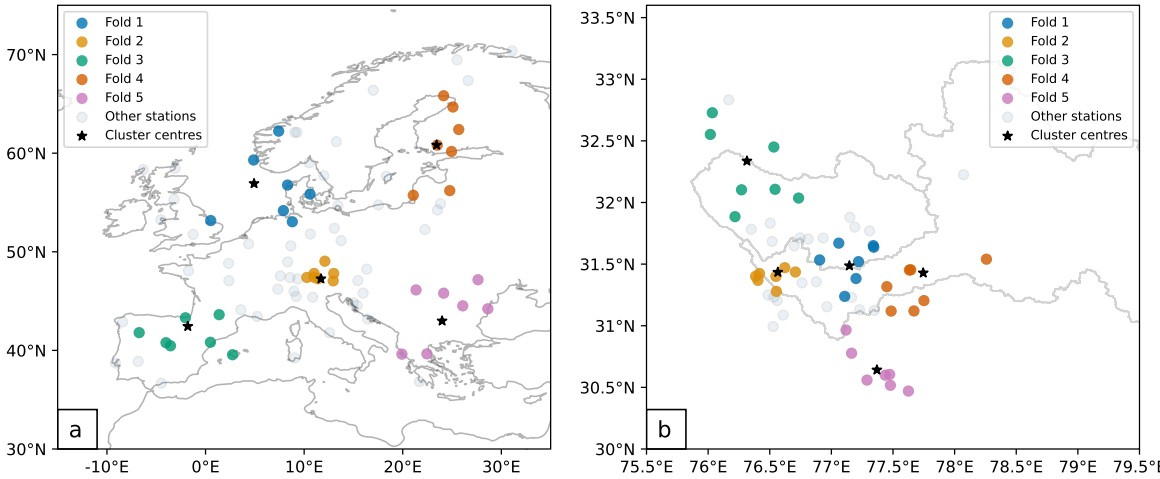

**Figure 3.** Maps of cross-validation folds over a) Europe and b) the upper Beas and Sutlej basins. The round marker represent the stations, the marker colours the different folds, and the stars the cluster centres found via k-means. The coastlines are plotted in black in (a) and the upper Beas and Sutlej basins watershed boundaries in light grey in (b).

low-fidelity ERA5 data:

$$
\tilde{y}_i = g_\lambda(y_i) = \begin{cases} \frac{y_i^\lambda - 1}{\lambda}, & \text{if} \quad \lambda \neq 0 \\ \log y_i, & \text{if} \quad \lambda = 0 \end{cases} \tag{7}
$$

where $y_i$ is the $i^{\text{th}}$ observation and is assumed to be positive, $\tilde{y}_i$ the transformed value, and $\lambda$ is the scaling factor. The input features are standardised by subtracting the mean and dividing standard deviation of the training set before they are passed to the models. This is also known as z-scoring and generally improves inference.

### 4.1.3 Kernel design

The MFGP kernels are specified to be Matérn ⁵⁄₂ functions defined as:

$$
\boldsymbol{k}_{\text{Mat}}(\boldsymbol{x}, \boldsymbol{x}') = \frac{\sigma^2}{\Gamma(\nu)2^{\nu-1}} \left( \frac{\sqrt{2\nu}}{l}|\boldsymbol{x} - \boldsymbol{x}'| \right)^\nu K_\nu \left( \frac{\sqrt{2\nu}}{l} \right) \tag{8}
$$

where $\nu = 5/2$, $\sigma^2$ is the variance parameter, $l$ the lengthscale parameter, $\Gamma$ is the gamma function, and $K_\nu$ is the modified Bessel function of the second kind. The Matérn ⁵⁄₂ function provides samples that are more faithful to real physical processes compared to the default squared exponential kernel. The samples are twice differentiable, i.e. not completely smooth, thus allowing for more abrupt changes in the modelled variable. The Matérn ⁵⁄₂ kernel performed better than the squared exponential kernel for both the Europe and upper Beas and Sutlej basin experiments (not shown).

### 4.1.4 Machine learning baselines

The performance of the MFGP is compared to several baseline models. We would like to establish that using both low and high-fidelity data improve models that use just one or the other. In order to do this, we implement a GP fit to ERA5 data using a Matérn ⁵⁄₂ kernel, and a GP fit to the station data using a Matérn ⁵⁄₂ kernel. The GP fit to ERA5 using a Matérn ⁵⁄₂ is equivalent to the MFGP low-fidelity output. Finally, the MFGP is also compared to a GP fit on the station data with the custom kernel design. The custom kernel is defined as:

$$k = k_{\text{Mat52}}(\text{time}) \cdot k_{\text{Per}}(\text{time}) + k_{\text{Mat52-ARD}}(\text{latitude, longitude, elevation}) \tag{9}$$

where $k_{\text{Mat52}}$ is the Matérn ⁵⁄₂ kernel, $k_{\text{Per}}$ is the periodic kernel, and $k_{\text{Mat52-ARD}}$ is the Matérn ⁵⁄₂ exponential kernel with Automatic Relevance Determination (ARD) (MacKay, 1994). ARD allows the kernel parameters to vary between input dimensions. The periodic kernel is defined as:

$$k_{\text{Per}}(\boldsymbol{x}, \boldsymbol{x}') = \sigma^2 \exp\left(-\frac{2\sin(2\pi|\boldsymbol{x} - \boldsymbol{x}'|/p)}{l^2}\right) \tag{10}$$

where $p$ is the period parameter, $\sigma^2$ is the variance parameter and $l$ the lengthscale parameter.[1] A similar kernel design to Equation 9 was used over the upper Indus basin with ERA5 precipitation by Lalchand et al. (2022) and was found to perform as well as more complex non-stationary kernel functions. The kernel design was formulated following the framework proposed Tazi et al. (2023) where statistical analysis of the precipitation data and domain knowledge, such as the periodic temporal patterns and the strong influence of elevation, were combined to create a kernel that is predictive without being unnecessarily complex.

Additionally, the MFGP model is compared to other models commonly used to interpolate or downscale precipitation for small datasets. We implement three non-probabilistic models including linear interpolation and random forest and support vector regression downscaling where ERA5 precipitation is directly used as a high-fidelity precipitation predictor. We also compare the MFGP with a strong alternative probabilistic model, namely a Convolutional Conditional Neural Process (ConvCNP). Although these models contextualise the MFGP performance, they do not contribute towards the main goal of demonstrating how the uncertainty can be narrowed by incorporating multiple data sources. For this reason, these models are discussed in Section 6.

### 4.1.5 Performance metrics

Several metrics are used to evaluate the models. The root mean square error (RMSE) is calculated for the validation sets as well as their 5th percentile and 95th percentile values to evaluate how well the model is capturing extremes. The RMSE is more robust to outliers than the mean absolute error or the bias. We also calculate the coefficient of determination ($R^2$) to understand how much of the variance in the data is represented by the model. These metrics are chosen in part for their broad usage across

---

[1]Although $\sigma^2$ and $l$ serve similar purposes to the parameters of the Matérn ⁵⁄₂ kernel shown in Equation 8, they are actually distinct variables and optimised separately.

both machine learning and environmental science fields. The mean log loss (MLL) computes average negative logarithm of the posterior likelihood of all validation points. This metric is a measure of the model confidence and the quality of its uncertainty predictions. The MLL is more suited to probabilistic methods than RMSE or $R^2$. All the metrics are defined in Appendix B.

## 4.2 Validation over Europe

The MFGP framework is first applied to a 'data-rich' setting over Europe. Table 1 shows the performance of the MFGP with respect to other simpler GP models. Of these methods, the GP with the custom kernel extrapolating only from gauges yields the poorest results with a negative $R^2$ indicating that the model is predicting worse than the precipitation mean. This poor predictive skill is expected as the custom kernel is designed to model precipitation over the Western Himalayas and not Europe. By contrast, precipitation estimates from the GPs with the Matérn kernels provide better results. In particular, applying a GP fit to ERA5 data at every station location gives even better estimates compared to a GP fit to the station data, including the best estimates for $95^{\text{th}}$ percentile RMSE (2.58±1.11 mm/day). However, the MFGP model gives the best overall results with lowest mean and $5^{\text{th}}$ percentile RMSE (1.06±0.42 mm/day and 0.51±0.20 mm/day respectively), the highest $R^2$ (0.65±0.09 mm/day), and the lowest MLL (0.89±0.20).

| Model | Training features | RMSE [mm/day] | RMSE5 [mm/day] | RMSE95 [mm/day] | $R^2$ | MLL |
|---|---|---|---|---|---|---|
| MFGP | gauges + ERA5 | **1.06±0.42** | **0.51±0.20** | 2.72±1.54 | **0.65±0.09** | **0.89±0.20** |
| GP$_{\text{Mat52}}$ | ERA5 | 1.16±0.43 | 0.52±0.25 | **2.58±1.11** | 0.57±0.13 | $(1.87\pm0.71)10^7$ |
| GP$_{\text{custom}}$ | gauges | 1.91±0.69 | 1.60±0.22 | 5.58±2.06 | -0.14±0.23 | 1.57±0.19 |
| GP$_{\text{Mat52}}$ | gauges | 1.21±0.45 | 0.59±0.29 | 2.85±1.17 | 0.55±0.14 | 1.94±0.36 |

**Table 1.** Comparison of model performance metrics for the 'data-rich' setup over Europe. The models include the MFGP, a GP using the custom kernel, and a GP using a Matérn ½ kernel with ARD. The metrics include the average RMSE (RMSE), the $5^{\text{th}}$ percentile RMSE (RMSE5), the $95^{\text{th}}$ percentile RMSE (RMSE95), the $R^2$, and the MLL. The training features represent inputs used to train the models. The errors represent the standard deviation across the validation folds. Bolded values show the best model performance for a given metric.

The experiment is then repeated for the 'data-sparse' setting. Table 2 shows the performance metrics for this setup. Despite a small decrease in performance compared to the 'data-rich' experiment shown in Table 1, the MFGP model is still able to combine the two datasets to improve predictions. The other baselines also show a generalised decreases in skill but their ranking is unaffected.

Figure 4 a) plots the high-fidelity output as a function of low-fidelity $R^2$ for the validation locations. The high-fidelity output corresponds to the MFGP fit using both ERA5 and the gauge data. The low-fidelity MFGP fit uses on ERA5 only and is equivalent to fitting a simple GP to ERA5 as shown in Table 2. Values above the dashed line show the locations where combining the datasets leads to improved performance. The plot shows that the MFGP improves predictions at most station locations. The largest gains are observed over the European alps (shown in orange). Simultaneously this is also the area, along

| Model | Training features | RMSE [mm/day] | RMSE5 [mm/day] | RMSE95 [mm/day] | $R^2$ | MLL |
|---|---|---|---|---|---|---|
| MFGP | gauges + ERA5 | **1.13±0.47** | **0.57±0.23** | 3.02±1.62 | **0.62±0.11** | **0.90±0.20** |
| $GP_{Mat52}$ | ERA5 | 1.21±0.46 | 0.59±0.29 | **2.84±1.17** | 0.55±0.14 | $(18.7±7.4) \, 10^6$ |
| $GP_{custom}$ | gauges | 2.25±0.90 | 1.10±0.60 | 6.51±2.29 | -0.57±0.46 | 1.73±0.33 |
| $GP_{Mat52}$ | gauges | 2.13±0.91 | 1.21±0.48 | 6.29±2.35 | -0.39±0.44 | 1.62±0.31 |

**Table 2.** As Table 1 for the 'data-sparse' setup over Europe.

with the Pyrenees and northern Spain (shown in green) where the model produces the largest errors. Altogether these results show that MFGPs can confidently be applied to more data-sparse locations.

### 4.3 Validation over upper Beas and Sutlej basins

Table 3 shows the performance of the MFGP with respect to other simpler GP models for the upper Beas and Sutlej basins. Overall the performance of the MFGP model and machine learning baselines is worse than over Europe, with all metrics showing a decrease in skill. This can be explained for two reasons. First, ERA5 is more accurate over Europe than the upper Beas and Sutlej basins (cf. Tables 2 and 3). Second, the precipitation in the High Mountain Asia presents more extreme seasonal variations, so is harder to predict (see Appendix C). The higher spatial heterogeneity of the precipitation over the upper Beas and Sutlej basins should not strongly contribute to the performance difference as the standardised spatial lengthscales between the European and Himalayan stations are similar (see Appendix C).

The MFGP's MLL and RMSE metrics suffer the most compared to the European experiments and the GP baselines. The MFGP's RMSE values grow approximately by a factor of 3 and the MLL by a factor of 2. This behaviour could be caused by the specific temporal distribution of precipitation in the upper Beas and Sutlej. For most of the year, precipitation values stay low but increase dramatically during the Indian Summer Monsoon, peaking in June/July. If the model does not predict these extreme values, the MLL and RMSE are heavily penalised. Conversely, the stronger periodicity in the data makes it easier to fit the GP models thus comparatively improving the GP MLL scores and 5[th] percentile RMSE. The MFGP still outperforms the GP fit to ERA5 and the GP extrapolating from the station data only with a mean RMSE ($3.00±0.92$ mm/day) and $R^2$ ($0.46±0.11$). In this experiment, the GP with the custom kernel outperforms the GP with the Matérn kernel suggesting that incorporating domain knowledge becomes more important in this more complex precipitation regime. The experiments were also conducted with all the ERA5 data for the study area (not shown), but showed no significant improvement over using the ERA5 data at the station locations only.

Figure 4 b) plots the high-fidelity $R^2$ as a function of low-fidelity $R^2$ for the validation locations across the basins. The figure shows that when the low-fidelity $R^2$ is already high (>0.5), the MFGP improvements are limited. However, when the low-fidelity $R^2$ is low, the MFGP significantly improves the low-fidelity fit. The upper Beas and Sutlej low-fidelity $R^2$ values also

| Model | Training features | RMSE [mm/day] | RMSE5 [mm/day] | RMSE95 [mm/day] | $R^2$ | MLL |
|---|---|---|---|---|---|---|
| MFGP | gauges + ERA5 | **3.00±0.92** | 1.66±0.95 | 9.62±3.63 | **0.46±0.11** | 1.79±0.22 |
| $GP_{Mat52}$ | ERA5 | 3.32±0.79 | 2.39±1.52 | **7.56±2.81** | 0.26±0.32 | $(11.4±4.8)\ 10^7$ |
| $GP_{custom}$ | gauges | 3.16±1.00 | 0.99±0.76 | 10.46±4.33 | 0.40±0.11 | **1.67±0.31** |
| $GP_{Mat52}$ | gauges | 3.24±1.35 | **0.86±0.56** | 11.0±5.11 | 0.38±0.25 | 1.66±0.32 |

**Table 3.** As Table 2 for the Upper Beas and Sutlej basins.

cover a much larger range. Although the MFGP improves the low-fidelity predictions less consistently than over Europe, it makes larger improvements over ERA5 over the upper Beas and Sutlej basins. In particular, the largest improvements are observed for Fold 4 (shown in red) which has the highest average elevation and is therefore most representative of the basins' 340 ungauged areas. This result is therefore encouraging given the paper's objective to predict in high altitude ungauged locations.

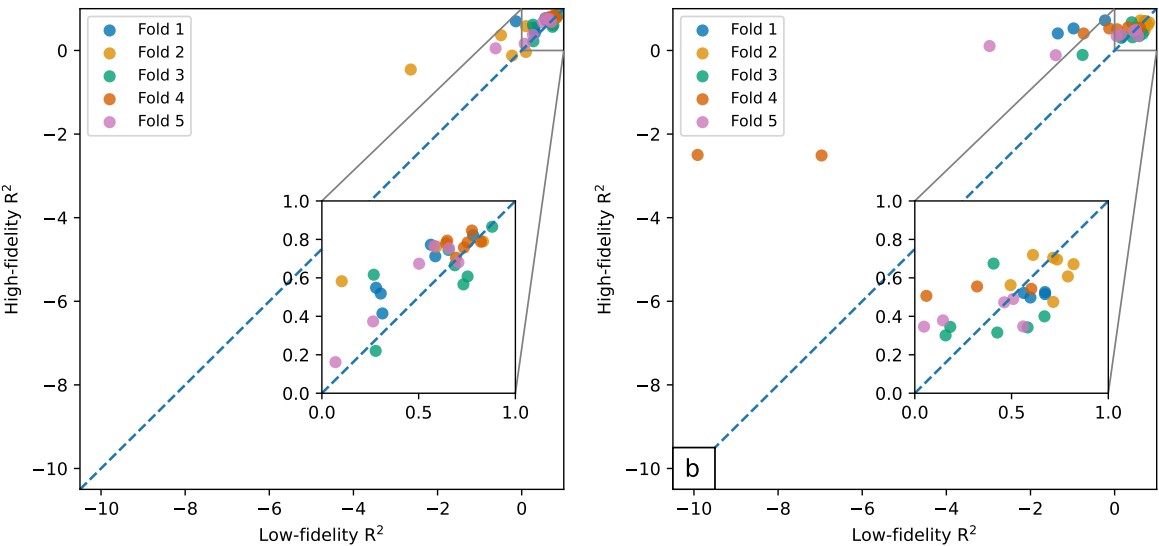

**Figure 4.** MFGP high-fidelity $R^2$ as a function low-fidelity $R^2$ over a) 'data-sparse' Europe and b) upper Beas and Sutlej basins. The colours correspond to the folds shown in Figure 3. Values above the dashed line show an improvement over the low-fidelity MFGP fit. The plots shows that as the low-fidelity $R^2$ decreases the high-fidelity $R^2$ stays relatively high. This illustrates that important gains can be made over using ERA5 alone.

## 5 Application to upper Beas and Sutlej basins

### 5.1 Study area predictions

A MFGP model is now trained using all available station data, including the stations outside of the basins, and ERA5 data over the study area (30° N-33.5° N, 75.5° E-83° E) between 2000 and 2009. This corresponds to the overlapping period between all the benchmark datasets studied in the following section. Again the precipitation values are transformed and input features are z-scored before they are passed to the model as this improves model inference. Separate models are trained on a yearly basis due to memory and computational constraints. In the Appendix D, we show that, assuming no missing data, this does not significantly impact the results of the model performance. When training the model across the entirety of both basins, the MFGP high-fidelity GP initially optimised the longitude lengthscale to a very small value. This produced nonphysical looking results with striations along lines of same longitude. Therefore, Gaussian prior distributions of $\mathcal{N}(0.1°, 0.01°)$ are set for the longitude and latitude lengthscale parameters such that they would optimise to similar values. This choice is motivated by the expectation that the precipitation lengthscales should be similar along these dimensions. The prior parameter values are selected based on the optimised hyperparameters for the MFGP's low-fidelity GP and the high-fidelity hyperparameters from the MFGP validation experiment. Finally, the GMTED2010 dataset was used (Danielson and Gesch, 2011) for the prediction locations and elevations. The dataset's 0.0625° resolution (approx. 7 km) allows the MFGP model to predict at high enough resolution to enable municipal decision making (Rambali, 2020).

The average annual and seasonal precipitation MFGP predictions are shown in Figure 5. The mean of the MFGP posterior distribution is compared to ERA5 precipitation in the first two rows. The MFGP annual average shows that most of the precipitation is concentrated in the west half of the study area over the Himalayan foothills. During the monsoon season, the MFGP shows an average rainfall reaching 10 mm/day. The monsoon also brings rain to the southeastern side of the upper Sutlej basin. Although estival precipitation distributions are similar, the highest precipitation values of the MFGP model are shifted west relative to ERA5. In the winter months, the variance in precipitation is more attenuated and the distribution centre is shifted to the North East and thus towards higher elevations. In contrast with the upper Beas basin, precipitation over the eastern upper Sutlej basin increases with altitude with valleys showing overall little rain or snowfall (<2 mm/day) across all seasons. These findings qualitatively echo previous studies on the spatiotemporal distribution of precipitation in this area, including the non-stationary and complex pattern of orographic precipitation gradients (Dahri et al., 2016; Bannister et al., 2019).

The 95% confidence interval (CI) of MFGP model is also plotted in Figure 5. This metric represents the interval in which 95% of the MFGP outputs fall into. The CI boundary values therefore show possible extreme precipitation values. The CI is therefore used as a measure of uncertainty. The most salient characteristic of the CI is that it is large in comparison to the mean of the posterior distribution, over 45 mm/day for several locations. For both the monsoon and winter seasons the CI is largest in the area around 32°N, 77°E at the western edge of the study area. This behaviour is linked to conflicting low and high-fidelity predictions where ERA5 suggests high precipitation values while the high-fidelity gauge data suggest the precipitation

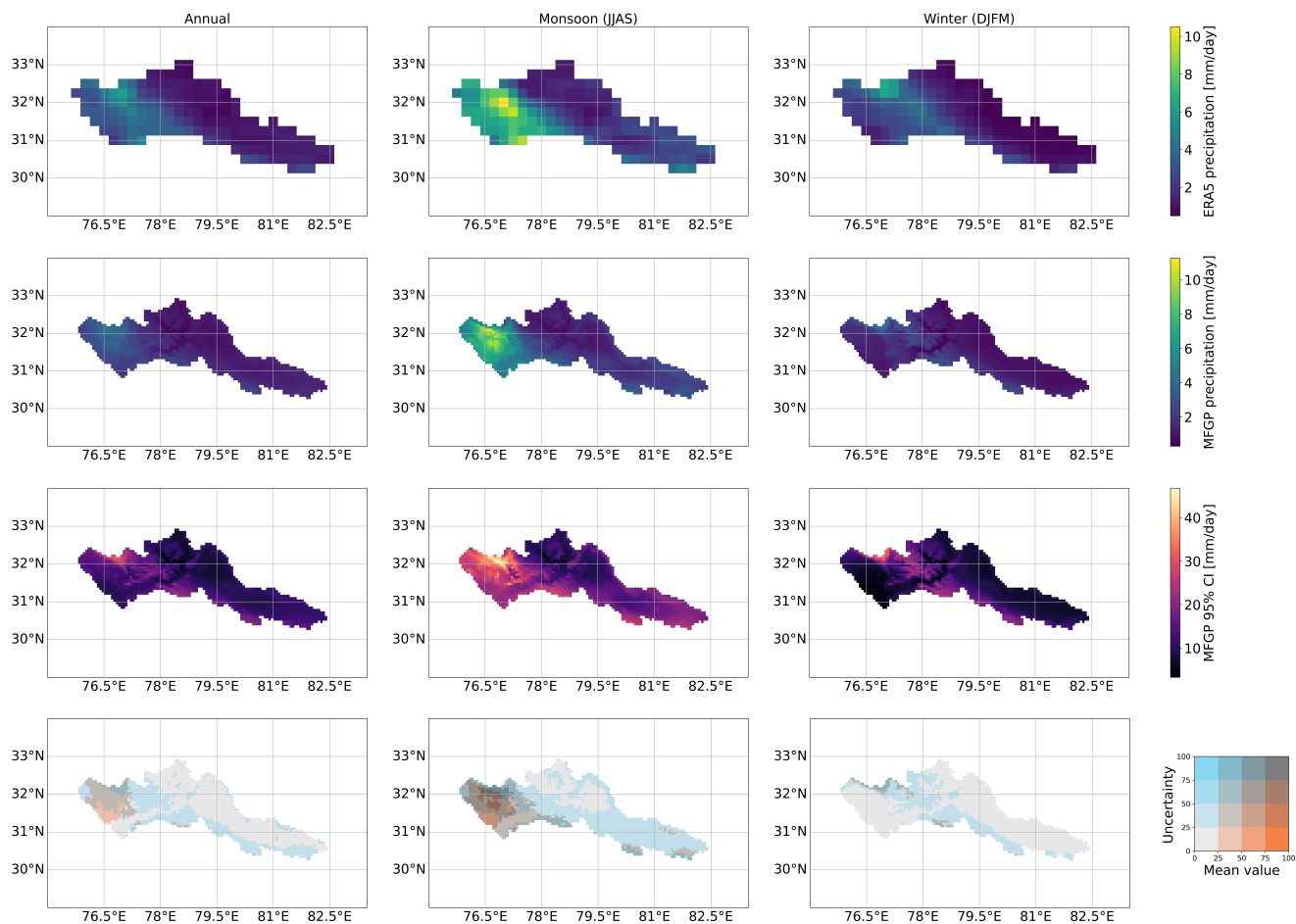

**Figure 5.** Spatial distribution of average precipitation from 2000 to 2009 over the upper Beas and Sutlej basins from ERA5 (top row), the MFGP posterior mean (second row), the MFGP 95% confidence interval (third row), and bivariate chloropleth map of the MFGP posterior mean and 95% confidence interval (bottom row). Here the 95% confidence interval is used as the measure of uncertainty. Results are shown for annual (first column), summer (JJAS; second column) and winter (DJFM; third column).

should be much smaller at the same location. Conversely, over ungauged areas, the CI remains low. This shows the improved predictive power of combining reanalysis and gauge data in a probabilistic framework.

The mean posterior distribution and CI are then combined in a bivariate chloropleth map in the bottom row of Figure 5. In general, the CI is expected to increase with higher precipitation values. This plot allows us to identify the regions that have the highest uncertainty output compared to their mean predictions, i.e. a high 'relative uncertainty'. The east and higher altitude ungauged locations generally have a high relative uncertainty, and areas with a high gauge density have a lower relative uncertainty. However, the chloropleth map does exhibit some smaller unexpected features. For example, a high relative uncertainty area in the west of the upper Beas basin (32°N, 76°E) and low relative uncertainty in the southern borders of the

upper Sutlej basin that receives more precipitation during the monsoon and winter seasons. Again, this points to the MFGP model successfully capitalising on information from both precipitation datasets.

The effective resolution of the MFGP model is also compared with that of ERA5. Effective resolution refers to the level of detail that can be accurately represented by the model. The effective resolution can be determined through the data's power spectrum. The power spectrum shows the amount of the structure present in the dataset for a given wavenumber $k$ or resolution $k^{-1}$. When the spectral density is low, it is not contributing structure at that resolution and therefore not representing the physical processes at that scale. To generate the power spectrum, a Fourier transform of the precipitation is calculated for each month over a square area (31°-33° N, 77°-79° E). To proceed equitably, the ERA5 data is linearly interpolated along its spatial coordinates to the same resolution as the MFGP and both datasets are z-scored. Figure 6 shows the spectral density $P$ falls off as a function of the resolution for both ERA5 and MFGP. Although ERA5 has a native resolution of 0.25° (approx. 31 km), it possesses a relatively small amount of structure compared to the MFGP the same resolution. The MFGP model continues to generate more structure at finer scales too. This points to the MFGP representing spatial patterns of precipitation better than ERA5.

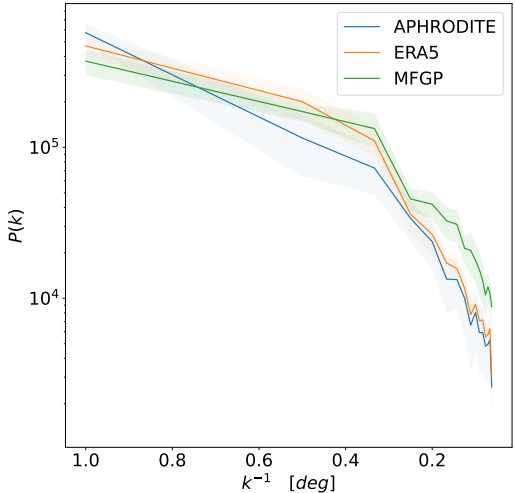

**Figure 6.** Power spectrum of the MFGP model, ERA5 and APHRODITE over a 2° by 2° area (31°-33°N, 77°-79°E) between 2000-2009. ERA5 and APHRODITE data are linearly interpolated to the same resolution as the MFGP output. The y-axis shows the power spectral density as a function of resolution, i.e. the inverse of the wavenumber $k$. The continuous lines show the average spectral densities and the shaded areas represent their standard deviation over time. All three datasets are z-scored prior to analysis.

## 5.2 Comparison with benchmark datasets

To further evaluate the performance of the MFGP model over the upper Beas and Sutlej basins, the benchmark datasets described in Section 3.3 are now are compared to the *in situ* observations between 2000 to 2009. All the available station data in the upper Beas and Sutlej basins (46 of 58 available stations) are used. Nearest neighbour precipitation values to the station

locations are reported. It is important to note that bias-corrected WRF has used these gauge measurements in its development. This is also most likely the case for APHRODITE and CRU TS. Table 4 compares the performance of the products across the different metrics. As the MFGP model is trained on all these datapoints, we do not include the model's performance here as to not make an inequitable comparison.

APHRODITE outperforms the other products for the mean RMSE (2.36±0.86 mm/day), $5^{th}$ percentile RMSE (0.56±0.61 mm/day) and $R^2$ (0.43±1.01) metrics. ERA5 has the best $95^{th}$ percentile RMSE (6.17±3.54 mm/day) but the poorest $5^{th}$ percentile RMSE (0.84±0.79 mm/day). For this area, TRMM simultaneously yields the worst mean RMSE (3.99±1.43 mm/day) and $95^{th}$ percentile RMSE (8.54±4.02 mm/day). The results for ERA5 and TRMM match previous findings, exhibiting wet and dry biases respectively (Kumar et al., 2021; Chen et al., 2021; Andermann et al., 2011; Shukla et al., 2019; Yin et al., 2008). The bias-corrected WRF product has the worst $R^2$ performance (-0.31±2.80). Overall the table shows that the performance of these models is highly heterogeneous across both basins with all metrics showing large standard deviations.

| | Input features | RMSE [mm/day] | RMSE5 [mm/day] | RMSE95 [mm/day] | $R^2$ |
|---|---|---|---|---|---|
| ERA5 | multiple[†] | 2.83±0.89 | 0.84±0.79 | **6.17±3.54** | -0.11±1.98 |
| APHRODITE | gauges | **2.36±0.86** | **0.56±0.61** | 6.45±3.46 | **0.43±1.01** |
| TRMM | remote sensing | 3.99±1.43 | 0.83±0.76 | 8.54±4.02 | -0.18±0.51 |
| CRU TS | gauges | 2.76±1.09 | 0.62±0.39 | 7.63±4.23 | 0.25±1.15 |
| Bias-corrected WRF | gauges + WRF | 3.13±0.94 | 0.73±0.92 | 7.02±3.34 | -0.31±2.80 |

**Table 4.** RMSE and $R^2$ values for benchmark datasets over Upper Beas and Sutlej Basins between 2000 and 2010. Only stations located in the basins (46 out 58) are used to evaluate the datasets. The errors represent the standard deviation across the stations. As some of these benchmarks are or are likely produced using the station data, it is not possible to compare these results with the previous table. Bolded values show the best model performance for a given metric. [†] ERA5 uses only remote sensing data for precipitation measurements but is also constrained using direct measurements for other climatic variables.

From Table 4, APHRODITE was determined to be the most accurate of the benchmarks presented in the paper for this region and time period. The differences between APHRODITE and the MFGP output are therefore compared. The average precipitation across the basin for the MFGP output and APHRODITE between 2000 and 2009 do not differ much, with a mean and standard deviation of 1.73 mm/day and 2.37 mm/day respectively for the MFGP model compared to 1.61 mm/day and 2.33 mm/day for APHRODITE. Figure 7 maps out the annual and seasonal averages. The annual average shows local spatial differences on the order of ±2.5 mm/day. However seasonal averages show a much larger differences between the two datasets. In particular, APHRODITE predicts lower precipitation values in the northwest corner of the upper Beas basin (-5 mm/day to -8 mm/day) and higher values on southeast side of the upper Beas basin (+2.5 mm/day to +5 mm/day) during the summer monsoon. These difference are large compared to the values shown in Figure 5. This is also where MFGP places the most uncertainty in Figure 5. These results, in combination with the spatial differences between the MFGP and ERA5, point to an

420 ambiguous spatial representation of peak precipitation values in the Himalayan foothills during the monsoon. In the winter, the differences are smaller due to on average lower precipitation rates (between +5 and -1.5 $\mathrm{mm/day}$). During this period, MFGP model predicts lower precipitation estimates at higher altitudes compared to APHRODITE. Finally, the power spectrum for APHRODITE is calculated in Figure 6. The dataset presents a smaller average effective resolution compared to the MFGP and even ERA5.

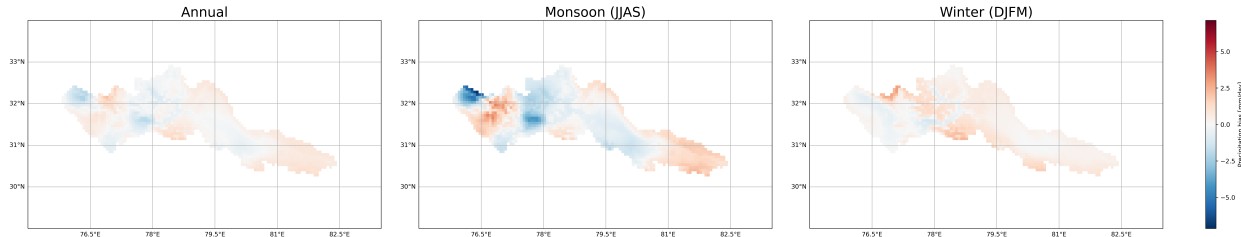

**Figure 7.** APHRODITE - MFGP differences between 2000 and 2009 over the upper Beas and Sutlej basins. Columns represent outputs for annual, monsoon (JJAS), and winter (DJFM) averages respectively.

## 6 Discussion and further work

### 6.1 MFGP extensions

The MFGP model is easily applicable to other watersheds and mountainous regions such as the Andes or European Alps, or to downscale other reanalysis or climate models. The model resolution is also arbitrary and higher resolution results could be generated by using a higher resolution digital elevation model when predicting at new times and locations. This flexibility
makes MFGP a powerful tool for hydrological and, more generally, geophysical modelling.

In this paper, a linear setup was applied. However, it is possible to apply the nonlinear form of the model, known as the Nonlinear Auto-Regressive GP (NARGP) (Perdikaris et al., 2017):

$$f_t(\boldsymbol{X}_t) = g_t\left(\boldsymbol{X}_t, f_{t-1}(\boldsymbol{X}_t)\right), \tag{11}$$

where $g_t \sim GP(f_t(\boldsymbol{X}_t)|\boldsymbol{0}, k_{t_g}((\boldsymbol{X}_t, f_{t-1}(\boldsymbol{X}_t)), (\boldsymbol{X}_t', f_{t-1}(\boldsymbol{X}_t'); \boldsymbol{\theta}_{kt}))))$. Unlike linear MFGP, NARGP captures a nonlinear
relationships between the different fidelities. However, the auto-regressive architecture of the model is also one of its limitations. The model specifies each GP is fitted in an isolated hierarchical manner. This type of inference means the model's complexity is not controlled through Bayesian inference and makes it more susceptible to overfitting. This was found to be true for the precipitation datasets presented in this paper. An alternative could be to implement a Multi-Fidelity Deep Gaussian Process (MFDGP) proposed by Cutajar et al. (2019) where the evaluation at each fidelity level is performed using data from
the current and previous fidelity levels. However, the MFDGP method requires the use of inducing points which can be hard to initialise without strong machine learning and environmental domain knowledge.

## 6.2 MFGP validation

### 6.2.1 With respect to GPs

In the validation experiments, we use datasets with no or a small number of missing values to compare the performance of the model with other methodological benchmarks. In this case, we are only evaluating how well the model extrapolates in space. This works in favour of the simple GP model that extrapolates from the high-fidelity gauge data. However, the simple GP's accuracy suffers significantly when extrapolating with respect to time, which is required when making predictions for incomplete datasets. This behaviour is another advantage of using a multi-fidelity model. The model validation in this study also highlights the impact of the observation scarcity to model accuracy. Tackling the impact of climate change on water scarcity in High Mountain Asia therefore requires more data sharing initiatives and consistent investment in weather station maintenance and deployment.

### 6.2.2 With respect to benchmarks

The benchmark datasets are compared on the validation folds in Appendix C. In this experiment, the MFGP model is able to outperform the other models on some metrics over the data held out from the model (see Section 4.3). In particular, the model still scores the best for $R^2$ (MFGP $R^2$= 0.46$\pm$0.11 vs average of $R^2$=0.00) despite most of these datasets being produced using these *in situ* observations. This shows that the underlying variations of data are being more accurately captured by the MFGP model, even if the amplitude of those variations are captured less precisely (higher RMSE scores). This lower precision makes sense as we expect the model to widen its posterior distribution at locations far from its training distribution. Furthermore the MFGP product, unlike previous ones, includes principled uncertainty estimates in the form of probability distributions. This can allow policymakers to understand the likelihood of worst case scenarios of drought or flooding. These uncertainty distributions can also be directly used to inform the placement of future sensors through multi-objective Bayesian optimisation (Daulton et al., 2021, 2020). The MFGP model outputs could for example be combined with distance from roads and trails as a proxy for accessibility. Together station locations that are both predictive and practical could be found. Finally, the model can be easily updated with new station data through online learning, a feature which is unique to Bayesian inference (Bui et al., 2017; Lederer et al., 2021).

### 6.2.3 With respect to other machine learning models

The performance of the MFGP model is also contextualised through the implementation of three non-probabilistic baseline models and a probabilistic deep learning model. Results and model implementation details are presented in Appendix E.

The performance of the linear interpolation model is first assessed. We note that the model presented in this paper is similar to the interpolation scheme used for precipitation in ERA5-Land (Muñoz-Sabater et al., 2021). ERA5-Land is a reanalysis dataset that provides a consistent view of the evolution of land variables at an enhanced spatial resolution of 9 km. This is produced by running a land surface model to regenerate some of the land components of ERA5 climate reanalysis. For atmospheric forcing,

it uses ERA5 atmospheric variables including precipitation which are linearly interpolated to the ERA5-Land grid. The linear interpolation model also includes elevation as a predictor which should allow it to perform better than ERA5-Land especially over mountainous regions.

Overall, linear interpolation performs significantly worse over both Europe and the Beas and Sutlej basins than the MFGP, and even its probabilistic counterpart, the GP fit to ERA5. This can be attributed to the GP's generation of non-linear functions that better capture ERA5's physics and data assimilation.

We then contrast the MFGP to random forest and support vector regression. Both random forests (Ho, 1995) and support vector regression (Drucker et al., 1996) have been used extensively to downscale precipitation, including over High Mountain Asia (Sun et al., 2022; Xiang et al., 2024; Ahmed et al., 2020; Yan et al., 2022; Ning et al., 2016; Mei et al., 2020). Both methods work well with small datasets, are non-linear, and, for support vector regression, are kernel-based like GPs.

The random forest and support vector regression models perform similarly to the MFGPs in terms of RMSE/$R^2$ for the 'data-rich' Europe experiment. However the MFGP performs consistently better for these metrics and is less sensitive to the reduction of data when moving to the 'data-sparse' setup. Over Europe, the random forests are however better at representing extreme values across all the cross validation folds. Over the Beas and Sutlej basins, the MFGP dominates offering more better and more consistent results with the exception of the 5th percentile RMSE. The relatively poor performance for the low percentiles values is due to the GP and MFGP models reverting to the observation mean in locations far from the training distribution where they are uncertain rather than confidently predicting lower values like the non-probabilistic models.

Lastly, ConvCNPs are also implemented for the validation experiments. The ConvCNP model is one member of the neural process model family that has shown state-of-the-art performance in spatiotemporal downscaling tasks (Vaughan et al., 2022; Gordon et al., 2019; Andersson et al., 2023). Neural processes offer similar advantages to the MFGP in terms of being able to quantify the probability of extreme events, generalise to multiple locations, predict at arbitrary locations, and overcome gridding biases. The results show that these models overfit these relatively small datasets performing worse than linear regression, in particular, for the Beas and Sutlej experiment. This is not surprising as neural networks generally require a large number of datapoints to be trained adequately. As these models can be used for transfer learning, future work could investigate the using data from other mountainous regions to inform predictions in data-sparse High Mountain Asia. In summary, the MFGPs are best suited to downscaling in the sparse and out-of-distribution settings presented in this paper.

### 6.3 Applicability of results

The MFGP model output for 33-year period between 1980 and 2012 over the upper Beas and Sutlej basins is made available for scientists, hydrologists and policymakers to perform more thorough research and water security risk assessments (Tazi, 2023). However, there are several limitations to its applicability. A key shortcoming to the results, as with many precipitation product in mountainous areas, is the underestimation of precipitation estimates due to undercatch. This is especially true in exposed areas and where precipitation falls as snow. Implementing the model a year at a time is also problematic. This means the model,

at times with fewer observations, cannot leverage the mappings that exist at other times. Furthermore, predictions have been made for a monthly resolution only and are inappropriate for hydrological models that usually operate on a daily timescale. These constraints come from the computational complexity of the MFGP. The framework could also be applied across High Mountain Asia but this would also be computationally expensive. These problems could be overcome by applying variational, product-of-experts, or low-rank approximations to the MFGP model (Tresp, 2000; Titsias, 2009; Wilson and Nickisch, 2015).

## 7   Conclusions

MFGPs are simpler and more accurate than recent state-of-the-art models and traditional techniques for smaller study areas with sparse datasets. The framework offers better mean RMSE and $R^2$ than the bias-corrected regional climate model output at prediction time. MFGP and APHRODITE perform similarly on average. Contrasting the two products across the basins shows general consensus about the total amount of annual precipitation. However, there are key areas where predictions diverge

including over high altitudes in the winter and the north of the upper Beas basin during the summer monsoon. Furthermore, the MFGP model also provides principled and well-calibrated uncertainty quantification. The model also provides a higher effective spatial resolution, providing more than three times the structure than ERA5 and APHRODITE at a 0.25° resolution. The continued improvements of these estimates will be key factors to improving hydrological modelling and water security policy. Future work could apply the framework across High Mountain Asia, predict precipitation on daily timescale, conduct

sensor placement analysis, and implement variational, product-of-experts or low-rank approximations to MFGP framework to improve computational tractability.

*Code and data availability.*   The code for this paper is available at: https://github.com/kenzaxtazi/mfgp. The MFGP model output between 1980 and 2012 for the upper Beas and Sutlej basin is available at: https://doi.org/10.5285/b2099787-b57c-44ae-bf42-0d46d9ec87cc. The ERA5 data is available through the Copernicus Data Store (https://cds.climate.copernicus.eu/). The VALUE gauge data is available through

the VALUE experiment website (http://www.value-cost.eu/data). The GMTED2010 elevation data used is available from the Tropospheric Emission Monitoring Internet Service (https://www.temis.nl/data/gmted2010/index.php). The authors of this paper do not have the required permission to make the Beas and Sutlej gauge dataset publicly available and suggest that any readers interested in obtaining it refer to Wulf et al. (2016).

## Appendix A: More Bayesian inference

### A1 Learning Gaussian Process hyperparameters

For multiple input-output pairs, $\boldsymbol{X}$ and $\boldsymbol{Y}$, the logarithm of the marginal likelihood is calculated. This is defined as the probability density of the observations given the hyperparameters:

$$\log(p(\boldsymbol{Y}|\boldsymbol{X},\boldsymbol{\theta})) = -\frac{1}{2}(\boldsymbol{Y}-\boldsymbol{\mu})^T(\boldsymbol{K}+\sigma_n^2\boldsymbol{I})^{-1}(\boldsymbol{Y}-\boldsymbol{\mu}) - \frac{1}{2}\log(|\boldsymbol{K}+\sigma_n^2\boldsymbol{I}|) - \frac{N}{2}\log(2\pi) \tag{A1}$$

where $\boldsymbol{K}$ is the covariance matrix constructed from the kernel function $k$, $\sigma_n$ is the noise specified at the observations. The logarithm of the likelihood is used to simplify the differentiation during Maximum Likelihood Estimation of the hyperparameters.

### A2 Predicting with Gaussian Processes

Assuming a Gaussian likelihood for $\epsilon$ (see Equation 1), calculating the posterior distribution $p(f_*|\boldsymbol{Y},\boldsymbol{X})$ is tractable and can be used to perform predictive inference for a new outputs $f_*$, given a new inputs $\boldsymbol{X}_*$ as:

$$p(f_*|\boldsymbol{Y},\boldsymbol{X},\boldsymbol{X}_*) = \mathcal{N}(f_*|\mu_*(\boldsymbol{X}_*),\sigma_*^2(\boldsymbol{X}_*)) \tag{A2}$$

Predictions are computed using the posterior mean $\mu_*$, while the uncertainty associated with these predictions is quantified through the posterior variance $\sigma_*^2$:

$$\mu_*(\boldsymbol{X}_*) = \boldsymbol{k}_{*N}(\boldsymbol{K}+\sigma_n^2\boldsymbol{I})^{-1}\boldsymbol{Y} \tag{A3}$$

$$\sigma_*^2(\boldsymbol{X}_*) = \boldsymbol{k}_{**} - \boldsymbol{k}_{*N}(\boldsymbol{K}+\sigma_n^2\boldsymbol{I})^{-1}\boldsymbol{k}_{*N}^T \tag{A4}$$

where $\boldsymbol{k}_{*N} = k(\boldsymbol{X}_*,\boldsymbol{X})$ and $\boldsymbol{k}_{**} = k(\boldsymbol{X}_*,\boldsymbol{X}_*)$. In other words, the variance captures how much uncertainty remains after seeing the data.

### A3 MFGP inference

At each level of the MFGP, the predicted mean $\mu_t$ and variance $\sigma_t^2$ are given by:

$$\mu_t(\boldsymbol{X}_*) = \rho_t\mu_{t-1}(\boldsymbol{X}_*) + \mu_{\text{err}} + \boldsymbol{k}_{*N_t}\boldsymbol{K}_t^{-1}[\boldsymbol{Y}_t - \rho_t\mu_{t-1}(\boldsymbol{X}_t) - \mu_{\text{err}}] \tag{A5}$$

$$\sigma_t^2(\boldsymbol{X}_*) = \rho_t^2\sigma_{t-1}^2(\boldsymbol{X}_*) + \boldsymbol{k}_{**} - \boldsymbol{k}_{*N_t}\boldsymbol{K}_t^{-1}\boldsymbol{k}_{*N_t}^T \tag{A6}$$

where $\boldsymbol{X}_*$ is a set of test points used over the domain of interest and $N_t$ denotes the number of training point locations where we have observed data from the $t^{\text{th}}$ information source. The mean and the uncertainty are thus elegantly propagated from one fidelity layer to the next. As the sum of two GPs is another GP, we can also write out the MFGP model as:

$$\begin{bmatrix} f_{t-1} \\ f_t \end{bmatrix} \sim GP\left(\begin{bmatrix} \mu_{t-1} \\ \mu_t \end{bmatrix} \begin{bmatrix} k_{t-1} & \rho_t k_{t-1} \\ \rho_t k_{t-1} & \rho_t^2 k_{t-1} + k_{\text{err}} \end{bmatrix}\right) \tag{A7}$$

## Appendix B: Metric definitions

### B1 Root mean square error

The RMSE represents the typical distance of the model from the data. For multiple input-output pairs $\{\boldsymbol{X}, \boldsymbol{Y}\}$, the RMSE is given by:

$$\text{RMSE} = \sqrt{\langle(\boldsymbol{Y} - \boldsymbol{f})^2\rangle} \tag{B1}$$

where and $\boldsymbol{f}$ the predicted value at $\boldsymbol{X}$. We use $\langle \cdot \rangle$ here and in the following definitions as a shorthand for the average over the datapoints. The RMSE is sensitive to outliers and systematic errors. The $5^{\text{th}}$ and $95^{\text{th}}$ percentile RMSE values are calculated by computing the RMSE for the high-fidelity data points in the $5^{\text{th}}$ and $95^{\text{th}}$ percentiles respectively.

### B2 Coefficient of determination

The $\text{R}^2$ represents the percentage of the data variance that can be explained by the model. It is given by:

$$\text{R}^2 = 1 - \frac{\text{SS}_{\text{res}}}{\text{SS}_{\text{tot}}} = 1 - \frac{\sum_i^N (y_i - f_i)^2}{\sum_i^N (y_i - \bar{y})^2} \tag{B2}$$

where $y_i$ is the $i^{\text{th}}$ observed value, $f_i$ the $i^{\text{th}}$ predicted value and $N$ the total number of datapoints. $\text{SS}_{\text{res}}$ is the sum of the squared residuals and $\text{SS}_{\text{tot}}$ the total sum of squares. A $\text{R}^2$ of 1 indicates a perfect fit whilst a negative $\text{R}^2$ means the model performs worse than the mean. Although negative $\text{R}^2$ scores are unlikely in interpolation settings, they are possible when making predictions outside of the training distribution.

### B3 Mean log loss

Using the predictive distribution at each test input, the probability of the target given the model can be calculated. The log loss (Rasmussen et al., 2006) is given by taking the negative logarithm of this probability. Taking the mean over all inputs gives the mean log loss (MLL):

$$\text{MLL} = -\langle \log p(\boldsymbol{Y}|\boldsymbol{\theta}, \boldsymbol{X})\rangle = \left\langle \frac{1}{2}\log(2\pi\boldsymbol{\sigma}^2) + \frac{(\boldsymbol{Y} - \boldsymbol{\mu})^2}{2\boldsymbol{\sigma}^2}\right\rangle \tag{B3}$$

where $\boldsymbol{\theta}$ are the optimised hyperparameters and $\boldsymbol{\sigma}$ and $\boldsymbol{\mu}$ are the predicted means and variance at $\boldsymbol{X}$. Smaller values imply more skill. The MLL is calculated prior to the inverse Box-Cox transformation, as this metric assumes the model output is Gaussian.

## Appendix C: Further data analysis

This appendix brings together more analysis around the validation experiments. More specifically, Table C1 compares observational data over Europe and the upper Beas and Sutlej basins and their optimised GP hyperparameters. Overall this breakdown shows that the distribution of precipitation over the upper Beas and Sutlej basins is more complicated than that over Europe despite a similar standardised gauge density/GP lengthscales between gauges.

| Metric | mean | std dev | 5th percent | 95th percent | GP $l_{lon}$ | | GP $l_{lat}$ | |
| --- | --- | --- | --- | --- | --- | --- | --- | --- |
| Unit | [mm/day] | [mm/day] | [mm/day] | [mm/day] | [°E] | z-scored | [°N] | z-scored |
| VALUE gauges | 2.39 | 2.19 | 0.22 | 6.61 | 4.96 | 0.47 | 3.80 | 0.49 |
| BS gauges | 2.95 | 3.98 | 0.00 | 11.17 | 0.26 | 0.52 | 0.23 | 0.48 |

**Table C1.** Precipitation statistics over Europe and the Beas and Sutlej using gauge data from 2000 to 2005. The mean, standard deviation, the 5th and 95th percentile values, and lengthscale values for the datasets are presented. The lengthscales are calculated by fitting a GP with a Matérn ⁵⁄₂ kernel to each of the gauge datasets with time, latitude, longitude and elevation as inputs.

Table C2 shows the performance of the benchmark datasets for the upper Beas and Sutlej validation experiment. These results are not directly comparable to the MFGP model as the data used to create these products are or are likely included in the held out validation sets. They can however give us a indication of how well these models perform in absolute terms for this gauged area.

| | Input features | RMSE [mm/day] | RMSE5 [mm/day] | RMSE95 [mm/day] | $R^2$ |
| --- | --- | --- | --- | --- | --- |
| ERA5 | multiple[†] | 3.03±1.05 | 0.53±0.58 | 6.01±3.52 | -0.30±2.18 |
| APHRODITE | gauges | **2.27±0.92** | **0.27±0.30** | **5.35±3.26** | **0.45±0.68** |
| TRMM | remote sensing | 3.83±1.36 | 0.58±0.77 | 8.18±4.27 | -0.22±0.68 |
| CRU TS | gauges | 2.87±1.20 | 0.43±0.24 | 7.59±4.71 | 0.19±1.22 |
| Bias-corrected WRF | gauges + WRF | 3.12±1.00 | 0.37±0.72 | 7.02±4.21 | -0.10±1.77 |

**Table C2.** RMSE and $R^2$ values for benchmark datasets over the Upper Beas and Sutlej Basins between 2000 and 2005 for cross-validation test stations. The errors represent the standard deviation across the cross validation folds. Bolded values show the best model performance for a given metric. [†] ERA5 uses only remote sensing data for precipitation measurements but is also constrained using direct measurements for other climatic variables.

## Appendix D:  MFGP time sensitivity

The computational complexity of the MFGP framework only allows the modeller to train over climatically short periods of time. In this study, we assume that long term variability is accurately captured by ERA5 and that there is limited information 590 to learn by training over longer time periods. This assumption is tested in the following experiment where we repeat the 'data-sparse' version of the European validation experiment over different time ranges. Figure D1 shows the model performance as a function of the number of time points for the different folds. Aside from a dip at the 2 year mark, there is no generalised trend change between different time periods across folds.

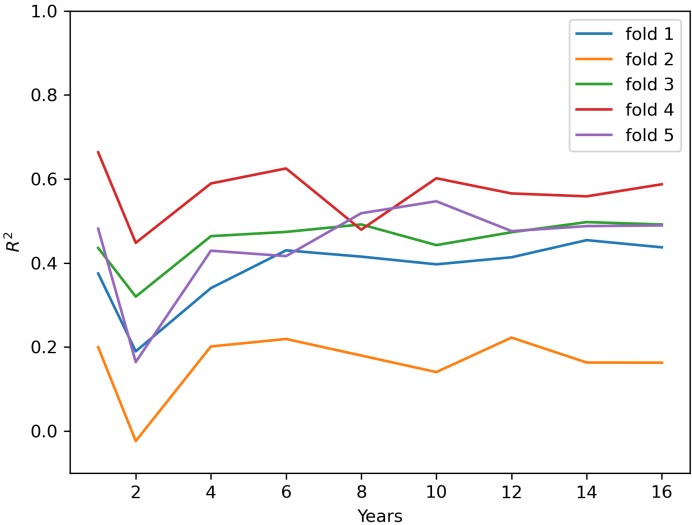

**Figure D1.** $R^2$ as function of years used to model the data across the different folds of the 'data-sparse' experiment over Europe.

# Appendix E: Machine learning baseline results

To contextualise the performance of the MFGP models linear interpolation and downscaling using random forests and support vector regression. The models are applied to the validation experiments presented in Section 4. These models have no explicit way of merging multiple data sources, instead we use ERA5 as a fifth input to models. The random forest models were trained with 100 trees and the stopping tolerance for the support vector regression model was set to $10^{-3}$. We note that no systematic hyperparameter search was performed for these models.

We also compare the MFGP to a Convolutional Conditional Neural Process (ConvCNP). In this setup, we used the high-fidelity elevation as a context dataset to the model. The model itself was trained using a U-Net with four downsampling layers each with 64 channels, an internal density of 500 and a learning rate of $5 \times 10^{-5}$ and sampling all the data at each timestep to create the training tasks. The models are trained for 20 and 15 epochs for the Europe and Beas-Sutlej experiments respectively. Again no systematic hyperparameter search or tailored sampling approach was performed for the ConvCNP models.

| Model | Training features | RMSE [mm/day] | RMSE5 [mm/day] | RMSE95 [mm/day] | $R^2$ | MLL |
|---|---|---|---|---|---|---|
| Linear reg. | ERA5 | 1.72±0.46 | 1.75±0.18 | 5.21±1.55 | 0.04±0.06 | - |
| RF | ERA5 + gauges | 1.12±0.44 | **0.45±0.19** | **2.62±0.93** | 0.61±0.09 | - |
| SVR$_{RBF}$ | ERA5 + gauges | 1.14±0.46 | 0.53±0.33 | 3.03±1.48 | 0.60±0.12 | - |
| MFGP | ERA5 + gauges | **1.06±0.42** | 0.51±0.20 | 2.72±1.54 | **0.65±0.09** | **0.89±0.20** |
| ConvCNP | ERA5 + gauges | 2.16±0.76 | 2.29±0.93 | 4.25±1.60 | -0.49±0.48 | 2.40±0.91 |

**Table E1.** Comparison of model performance metrics trained on ERA5 data for the 'data-rich' setup over Europe. We include a linear interpolation model, a random forest (RF), a support vector regression (SVR) model with a smooth Radial Basis Function (RBF) kernel, a ConvCNP and the MFGP model. The metrics include the average RMSE, the 5[th] percentile RMSE (RMSE5), the 95[th] percentile RMSE (RMSE95), the $R^2$ score, and the MLL. The errors represent the standard deviation across the validation folds.

| Model | Training features | RMSE [mm/day] | RMSE5 [mm/day] | RMSE95 [mm/day] | $R^2$ | MLL |
|---|---|---|---|---|---|---|
| Linear reg. | ERA5 | 1.77±0.46 | 1.88±0.25 | 5.19±1.76 | -0.02±0.13 | - |
| RF | ERA5 + gauges | 1.16±0.39 | **0.41±0.20** | **2.92±1.39** | 0.57±0.10 | - |
| SVR$_{RBF}$ | ERA5 + gauges | 1.53±0.62 | 0.73±0.23 | 4.64±1.99 | 0.29±0.19 | - |
| MFGP | ERA5 + gauges | **1.13±0.47** | 0.57±0.23 | 3.02±1.62 | **0.62±0.11** | **0.90±0.20** |
| ConvCNP | ERA5 + gauges | 1.92±0.51 | 1.77±0.78 | 4.84±1.70 | -0.21±0.34 | 2.36±1.38 |

**Table E2.** As Table E1 for the 'data-sparse' setup over Europe.

| Model | Training features | RMSE [mm/day] | RMSE5 [mm/day] | RMSE95 [mm/day] | $R^2$ | MLL |
|---|---|---|---|---|---|---|
| Linear reg. | ERA5 | 4.21±0.99 | 2.21±0.45 | 14.33±4.03 | -0.08±0.05 | - |
| RF | ERA5 + gauges | 3.05±1.30 | **0.52±0.46** | 9.87±5.47 | 0.45±0.23 | - |
| SVR$_{RBF}$ | ERA5 + gauges | 3.36±1.66 | 0.66±0.38 | 11.05±6.14 | 0.34±0.33 | - |
| MFGP | ERA5 + gauges | **3.00±0.92** | 1.66±0.95 | **9.62±3.63** | **0.46±0.11** | **1.79±0.22** |
| ConvCNP | ERA5 + gauges | 4.89±0.93 | 3.57±0.79 | 14.16±3.85 | -0.51±0.32 | 3.95±0.88 |

**Table E3.** As Table E2 for the upper Beas and Sutlej basins.

*Author contributions.* KT designed the experiments, conducted the research, and wrote the manuscript under the supervision of AO, JGH, RET and SH. AO advised on the environmental aspects of the research and analysis. RET and JGH provided feedback on the machine learning experiments and final model design and evaluation. All authors contributed to the manuscript's clarity and coherence and approved the final submitted draft.

*Competing interests.* No competing interest are present.

*Acknowledgements.* This work was supported by the UK Engineering and Physical Sciences Research Council [award number 2270379]. The authors thank Tony Phillips for providing the watershed shapefiles.

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
