# Peer review of "Downscaling precipitation over High Mountain Asia using Multi-Fidelity Gaussian Processes: Improved estimates from ERA5"

_EGUsphere, 2023_

## Author Response (AR1)

**Author response to 'Downscaling precipitation over High Mountain Asia using Multi-Fidelity Gaussian Processes: Improved estimates from ERA5'**

Kenza Tazi, Andrew Orr, Javier Hernandez-González, Scott Hosking, Richard E. Turner

August 2, 2024

We thank both reviewers for their feedback on the manuscript. We discuss their comments (italicised) below.

**Response to Reviewer 1**

We thank Reviewer 1 for their thorough and positive feedback (e.g., "*Overall, this is one of the best scientific papers I have ever read, especially considering its highly technical nature*"). We appreciate their encouraging comments on the manuscript's presentation, including the clarity and conciseness of the writing, the precision of the mathematical formulae and the efficacy of the figures (e.g., "*All aspects of the paper appear to be very carefully prepared, including citation of relevant literature, readability of the text by a broad audience, presentation of methods and results, and technical precision in presentation of the mathematics and statistics*"). We are also grateful they recognised our efforts towards contextualising this research within existing literature and giving a balanced discussion about the advantages and disadvantages of the proposed method.

**Response to Reviewer 2**

We thank Reviewer 2 for their insightful comments and suggestions. We address the reviewer concerns point by point below.

**Comment 1**

"*First, ECMWF also provides high-resolution reanalysis precipitation data (ERA5 Land, hourly, 0.1 degree, 9 km), which is not considered in the manuscript. How does the generated MFGP precipitation estimates compare with ERA5 Land precipitation data?*"

ERA5-Land is a reanalysis dataset that provides a consistent view of the evolution of land variables at an enhanced spatial resolution of 0.1° by 0.1° (approx. 9km) compared to ERA5's resolution of 0.25° by 0.25° (approx. 31km). It is produced by running a land surface model to regenerate some of the land components of ERA5 climate reanalysis. For atmospheric forcing, it uses ERA5 atmospheric variables such as air temperature and precipitation at a 0.1° resolution by linearly interpolating the driving variables to the ERA5-Land grid. Although other forcing variables are corrected, this is not the case for precipitation. For further details please see Muñoz-Sabater et al. (2021) and ECMWF (2024). Precipitation characteristics from ERA5-Land are therefore very similar to ERA5 (Gomis-Cebolla et al., 2023; Xu et al., 2022; Xin et al., 2022). They also should theoretically perform worse than the linear regression models presented in our paper (Table 1-3) which also include elevation as a predictor.

Moreover, we would like to stress that we have carefully chosen four gridded precipitation datasets for the High Mountain Asia region, based on existing literature, to evaluate our model against. These are:

- APHRODITE, which is a gridded rain-gauge interpolated dataset for Asia considered the gold-standard for precipitation in High Mountain Asia,

- CRU TS, which is a global gridded rain-gauge interpolated dataset.

- A bias-corrected high-resolution regional climate model simulation, which used the Weather Research and Forecast (WRF) model at a spatial resolution of 5 km, with precipitation output corrected using local rain-gauge data for the region investigated in this manuscript.

- TRMM, which is a satellite-based precipitation dataset, designed to improve our understanding of precipitation in the current climate.

To address the reviewer's concern, we make the connection between ERA5-Land and our linear regression models clearer in a new subsection of the discussion:

"We note that the model presented in this paper is similar to the interpolation scheme used for precipitation in ERA5-Land (Muñoz-Sabater et al., 2021). ERA5-Land is a reanalysis dataset that provides a consistent view of the evolution of land variables at an enhanced spatial resolution of 9 km. This is produced by running a land surface model to regenerate some of the land components of ERA5 climate reanalysis. For atmospheric forcing, it uses ERA5 atmospheric variables including precipitation which are linearly interpolated to the ERA5-Land grid. The linear interpolation model also includes elevation as a predictor which should allow it to perform better than ERA5-Land especially over mountainous regions."

– Section 6.3.3

**Comment 2**

"*Second, the authors only consider a very simple machine learning model, i.e., linear regression, and complex deep learning models that require a lot of training data, including Convolutional Conditional Neural Processes (ConvCNP) and Convolutional Gaussian Neural Processes (ConvGNP). They neglect simple machine learning methods that do not need many data, such as random forest and support vector machines.*"

The main goal of the paper was to show that the precipitation uncertainty could be narrowed by combining datasets from multiple sources. This information allows hydrologists to quantify the probabilities of extreme events and policymakers to make better decisions with limited resources as highlighted in Section 1. However, we appreciate that the case for the performance of Gaussian processes could be better contextualised by including these models.

To address the reviewer's concern we implement random forests and support vector regression for the validation experiments and now include the performance of ConvCNPs. These results are shown below in Tables A-C and are presented in the manuscript under Appendix E with the linear regression model and the models' implementation details. A summary of the results and the rationale for choosing these models are included in the discussion:

[revised manuscript text omitted]

Table C: As Table B for the upper Beas and Sutlej basins.

**Comment 3**

"Finally, in the model comparison shown in Table 1-3, GP is only trained on ERA5 data at station locations. Why not use all ERA5 data in the study region to train and test the model? I would expect better model performance even for GPs."

The model has information about ERA5 at all the training and test locations for the validation experiments. These locations fall within the ERA5 grid boxes, so there is theoretically little to no additional information to be gained by including neighbouring grid box values. To check this, we ran an experiment that was trained on all ERA5 data for the Beas and Sutlej basins.

Results from this experiment are shown in Table D (below), and confirm that there is no added benefit in including this data. However, we have clarified this in the revised manuscript by adding additional text which states:

"The experiments were also conducted with all the ERA5 data for the study area (not shown), but showed no significant improvement over using the ERA5 data at the station locations only".

– Section 4.2

We note that we did not rerun these experiments over Europe, as we would have needed to apply methodological approximations to overcome the memory and computational bottlenecks that comes with this larger domain.

| Model | RMSE [mm/day] | RMSE5 [mm/day] | RMSE95 [mm/day] | $R^2$ | MLL |
|---|---|---|---|---|---|
| $MFGP_{limited}$ | **3.00±0.92** | 1.66±0.95 | **9.62±3.63** | **0.46±0.11** | 1.79±0.22 |
| $MFGP_{all}$ | 5.16±2.51 | **0.84±0.56** | 19.48±9.79 | 0.32±0.27 | **1.68±0.34** |

Table D: Comparison of MFGP performance using ERA5 for the whole study area (all) and using only ERA5 at the training and test site locations (limited) over Upper Beas and Sutlej Basins. The metrics include the average RMSE, the 5$^{th}$ percentile RMSE (RMSE5), the 95$^{th}$ percentile RMSE (RMSE95), the $R^2$ score and the mean log loss (MLL).The bolded values highlight the best scores

**Comment 4**

*"I believe the authors use the Nash-Sutcliffe efficiency (NSE) in the manuscript, rather than R2, which should always be non-negative values."*

We confirm that we are using the coefficient of determination or $R^2$ score. This metric, defined and explained in Appendix B, and is given by:

$$R^2 = 1 - \frac{SS_{res}}{SS_{tot}} = 1 - \frac{\sum_i (y_i - f_i)^2}{\sum_i (y_i - \overline{y})^2}$$

where $f_i$ is the $i^{th}$ predicted value, $y_i$ is the $i^{th}$ observed value, and $\overline{y}$ is the mean of the observations. $SS_{res}$ is therefore the sum of the squared residuals and $SS_{tot}$ is the total sum of squares. A negative $R^2$ is possible and would indicate that the model is predicting worse than the precipitation mean. Although negative $R^2$ scores are unlikely in interpolation settings, they are possible when making predictions outside of the training distribution. To address the reviewer's concerns, we make this interpretation of negative $R^2$ clearer in the main body of the paper and the appendix:

"Of these methods, the GP with the custom kernel extrapolating only from gauges yields the poorest results with a negative $R^2$ indicating that the model is predicting worse than the precipitation mean."

– Section 4.2

*"An $R^2$ of 1 indicates a perfect fit whilst a negative $R^2$ means the model performs worse than the mean. Although negative $R^2$ scores are unlikely in interpolation settings, they are possible when making predictions outside of the training distribution."*

– Appendix B

**Summary of changes**

- Make the connection between ERA5-Land and the linear regression model clearer in a new subsection of the discussion (Section 6.3.3).

- Implement random forests and support vector regression for the validation experiments and which we present in Appendix E. The results are discussed alongside the linear regression and ConvCNP performances in Section 6.3.3.

- Clarify the limited benefit of using additional ERA5 data for the validation experiments in Section 4.2.

- Make the interpretation of negative $R^2$ clearer in the main body of the paper and Appendix B.